# Genetic mapping in the red mason bee implicates *ANTSR* as an ancient sex-determining locus in bees and ants

Tilman Rönneburg[1], Demetris Taliadoros[1], Anna Olsson[1], Sara Magnusson[1], Linn Huser[1], Muhammad Nafiz Ikhwan Bin Nor Fuad[1], Turid Everitt[1], Giselle C. Martín-Hernández[1], Panagiotis Theodorou[2], Björn Cederberg[3], Robert J. Paxton[2], Karsten Seidelmann[4], Matthew T. Webster ⓘ[1]*

1 Department of Medical Microbiology and Biochemistry, SciLifeLab, Uppsala University, Uppsala, Sweden, 2 General Zoology, Institute of Biology, Martin Luther University Halle-Wittenberg, Halle (Saale), Germany, 3 Swedish Species Information Centre, Swedish University of Agricultural Sciences, Uppsala, Sweden, 4 Animal Physiology, Institute of Biology, Martin Luther University Halle-Wittenberg, Halle (Saale), Germany

* matthew.webster@imbim.uu.se

## Abstract

Haplodiploid inheritance, in which females are diploid and males are haploid, is found in all species of Hymenoptera. Sex in haplodiploids is commonly determined by the alleles present at a complementary sex determination (CSD) locus, with heterozygosity triggering the female developmental pathway. The identity of this locus differs among taxa and is only known in a few species. Here, we map a single CSD locus to a 2 kbp region in the genome of the red mason bee *Osmia bicornis.* It overlaps the long noncoding RNA *ANTSR*, which has been identified as the sex-determining gene in the invasive ant *Linepithema humile.* This locus is homozygous in diploid males and exhibits extremely high levels of haplotype diversity, consistent with the action of frequency-dependent selection. The elevated levels of heterozygosity in the CSD locus enable us to fine-map potentially functional genetic variation within it. We also identify elevated levels of genetic diversity in the ortholog of the CSD locus in five other bee and ant genera, suggesting that it may govern sex determination widely in Hymenoptera. Our data are consistent with the hypothesis that *ANTSR* evolved a role in sex determination over 150 million years ago and is the ancestral sex-determination locus of bees and ants.

## Introduction

A huge variety of genetic sex determination systems exists in nature [1,2]. The molecular pathways involved in determining sex are highly conserved among animal and plant taxa, but the molecular switches that provide the initial signal to specify sex are highly diverse. Haplodiploidy is a common mode of sex determination, in which

**Data availability statement:** All scripts and code associated with the project are available on GitHub (https://github.com/troe27/osmia-sex-locus). Illumina population sequencing data for *O. bicornis* are available at NCBI BioProject PRJEB88603. Data from *Anthophora* spp., *Bombus* spp., and *A. mellifera* are available at NCBI BioProjects PRJNA1255204, PRJNA646847, and PRJNA357367, respectively. Data from *L. pauxillum* and *L. albipes* are available at NCBI BioProjects PRJNA1330011 and PRJNA413432, respectively. Data for the *Formica* species are at NCBI BioProject PRJEB51899, and *S. invicta* at PRJNA685290. Spreadsheets of the contents of each nesting tube and cocoon mass are available as Supplementary Tables. Data underlying the figures as well as a snapshot of the code can be found on Zenodo (https://doi.org/10.5281/zenodo.17424723).

**Funding:** This study was funded by a grant from Erik Philip-Sörensens Stiftelse (2020) to M.T.W. No funders had any role in study design, data collection and analysis, decision to publish, or preparation of the manuscript.

**Competing interests:** The authors have declared that no competing interests exist.

**Abbreviations:** CSD, complementary sex determination; ROH, runs of homozygosity.

females are diploid and males are haploid. Up to 15% of arthropods have this mode of sex determination, including all Hymenoptera (wasps, bees, ants, and sawflies) [3]. However, the molecular mechanisms that trigger the molecular cascade translating ploidy differences into sex differences are only known in a few haplodiploid species.

The most widespread mechanism of sex-determination in Hymenoptera is inferred to be complementary sex determination (CSD). This system is likely to be the ancestral state in Hymenoptera [4,5] although other mechanisms are responsible in some taxa. For example, in the parasitoid wasp, *Nasonia vitripennis*, sex is determined by different patterns of imprinting of a single gene in haploids and diploids [6]. Under CSD, on the other hand, sex is specified by the alleles present at one or more genetic loci at which heterozygotes become female and hemizygotes become male [7].

So far, CSD loci have been fully characterized in one bee and one ant species. In the honeybee *Apis mellifera*, the CSD locus is a single protein-coding gene *complementary sex determiner (csd)* [8]. It has been shown that the presence of amino acid differences between two alleles of the Csd protein are necessary for its activation and to trigger female development [9]. In the Argentine ant *Linepithema humile* a single CSD locus has been mapped to a region overlapping the lncRNA gene *ANTSR* [10] (Fig 1A). It was demonstrated that heterozygosity at this locus leads to elevated expression of *ANTSR* which triggers the molecular cascade leading to female development. Both these molecular switches act by inducing alternative splicing of an ortholog of *transformer* (*tra*), which in turn interacts with *doublesex* (*dsx*) leading to female development. These two genes are highly conserved components of the sex determination pathway of insects [11].

In species with CSD, occasional homozygosity at one or more CSD loci results in diploid males [4,7] (Fig 1B). This is a relatively rare and deleterious outcome and diploid males are typically sterile. The fact that diploid males must be homozygous at CSD loci, whereas diploid females must have two different haplotypes, can be used to identify their location in the genome [8]. Furthermore, frequency-dependent selection at CSD loci results in the maintenance of a large number of haplotypes and minimizes the chance of inheritance of two copies of the same haplotype in diploids [12]. This leads to elevated levels of variation at CSD loci compared to the rest of the genome, also facilitating genome localization of CSD.

Bees and ants share a common ancestor ~153 million years ago [13,14]. We know little about the diversity of sex-determination loci within these groups (Fig 1A). Analysis of species related to *A. mellifera* has revealed the presence of diverse haplotypes in the *csd* ortholog in other members of the *Apis* genus [15]. However, an ortholog of honeybee *csd* has not been identified in members of the closely related genera *Bombus* and *Melipona* [16]. This indicates that the *csd* locus is likely a recent innovation in honeybees. The mechanism of sex determination in almost all of the >20,000 bee species is therefore unknown.

Here we genetically map the CSD locus in the red mason bee *Osmia bicornis* using freely mating populations. This species belongs to the insect family Megachilidae, which is the most closely related to the Apidae (including the genera *Apis*, *Bombus*,

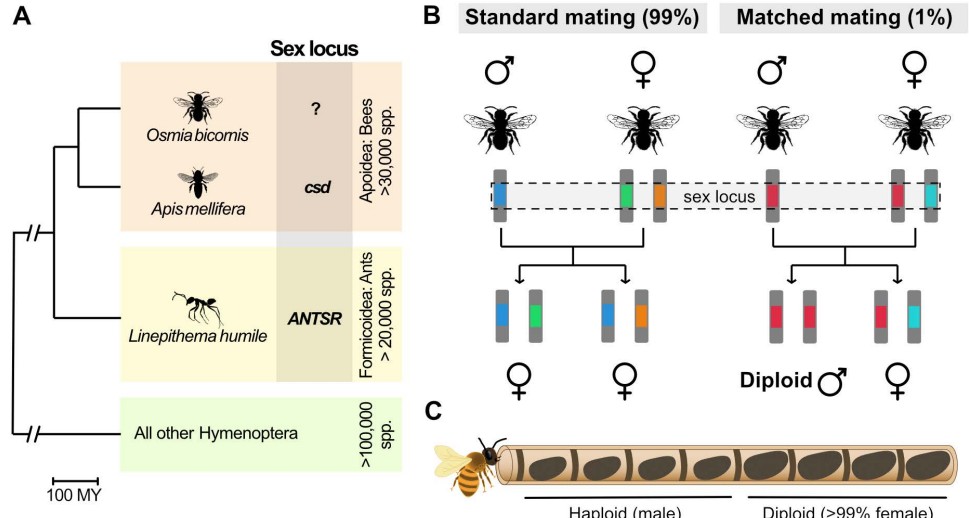

**Fig 1. Complementary sex determination (CSD) in bees and ants. A.** Single-locus CSD is likely to be the predominant mode of sex determination in Hymenoptera. CSD loci have been identified in two species: the honeybee *Apis mellifera* and the invasive ant *Linepithema humile*. The tree shows evolutionary relationships between these species and *Osmia bicornis*, the main subject of this study. The origin of Hymenoptera is estimated as 386 million years ago and the common ancestor of Apoidea and Formicoidea is estimated as 153 million years ago [14]. **B.** Under single-locus CSD, females are heterozygous at the CSD locus, whereas males are haploid and therefore hemizygous. A large number of CSD alleles are expected to segregate in outbred populations. A small proportion of homozygotes at the CSD locus are produced by chance, when parents share an allele, which results in the production of sterile diploid males. **C.** An *O. bicornis* nest consists of a row of cocoons in a tube. The mother lays fertilized diploid eggs deepest in the nest and unfertilized haploid eggs closer to the entrance. Eggs develop into adults inside cocoons. Diploid eggs typically develop into females whereas haploid eggs become male. A small proportion of diploids that are homozygous at the CSD locus develop into diploid males, which occur amongst females and are larger than haploid males. The silhouettes for *Osmia bicornis*, *Apis mellifera,* and *Linepitherma humile* were obtained from PhyloPic (https://www.phylopic.org, T. Michael Keesey, 2025) and were contributed by Thomas Hegna (2012; Public Domain Mark 1.0 Universal), Melissa Broussard (2023; Attribution 4.0 International), and Jitte Groothuis (2022; Attribution 4.0 International).

and *Melipona*) [13,14,17,18]. We phenotypically characterized bees from nesting tubes to identify potential diploid males. We then used whole-genome sequencing to identify the sex-determination locus based on homozygosity in diploid males and elevated genetic variation. Finally, we analyze genome-wide variation in heterozygosity in additional genera of bees (*Bombus*, *Anthophora*, *Lasioglossum*) and ants (*Formica* and *Solenopsis*) in order to search for CSD loci in these taxa.

## Results

### Identification of diploid males in natural populations of *O. bicornis*

We morphologically characterized *O. bicornis* individuals collected from 683 nesting tubes from two locations in Europe (Uppsala, Sweden and Halle, Germany). *O. bicornis* nests typically contain males closest to the entrance and females at the back, with males being lower in mass (Fig 1C). Potential diploid males were identified based on their occurrence among females in nests or due to elevated mass, or both (see Materials and methods). We also collected females from nests containing potential diploid males, and from randomly selected nests. In total, we performed whole-genome sequencing of 162 individuals (S1–S3 Tables). These comprised 88 potential diploid males and 74 females.

We mapped sequencing reads to the *O. bicornis* reference assembly iOsmBic2.1 to an average sequencing depth of 13×. We masked heterochromatic regions from the dataset (see Materials and methods, S1A and S1B Fig; S1 Table), resulting in a dataset consisting of 3,573,693 SNPs. We determined the ploidy of our samples by the presence of a significant number of heterozygous sites based on the inbreeding coefficient, *F,* which is expected to equal one in haploids (when analyzed in the same way as diploids). All females were confirmed as diploid. Out of the 88 potential diploid males,

18 were confirmed to be diploid, whereas 70 were haploid. Our final dataset, therefore, contained 18 diploid males, 70 haploid males, and 74 females (S1C and S1D Fig).

Females from nests containing diploid males had a slight but significantly higher $F$ than those sampled from random nests (mean $F = 0.452$ compared to 0.230, Mann–Whitney $U$ test, $p = 0.0042$). However, we found no significant difference in $F$ between diploid males and their sisters (mean $F = 0.298$ compared to 0.452, $p = 0.356$). Overall, this indicates that matings that produce diploid males are between individuals that are more closely related than average, but are not close relatives.

After determining the ploidy of all potential diploid males, we were able to estimate the frequency of matched matings in the population in which the mother shares an allele at the CSD locus with the father, leading to diploid male production. We corrected for the probability that a diploid male is not observed in a matched mating based on the number of progeny in each nest. In total, we observed 6 nests containing diploid males from a total of 672 nests found in the nesting tubes. We estimate a matched mating frequency of 1.04% (SD 0.44%; S4 Table). This frequency did not differ significantly between locations or sampling year (Fisher's exact test, $p > 0.05$ for all comparisons). The mass of confirmed diploid males was not statistically different from female nestmates, or from all females (S2A Fig; $t$ test, $p > 0.05$).

## Candidate CSD locus orthologous to *ANTSR* identified on chromosome 1

We next analyzed the genomes of diploid males and females for runs of homozygosity (ROH). The percentage of the genome in ROH blocks >10 kbp ranged from 0.19% to 67.4% between samples (S2B Fig). The distribution of this proportion was similar between diploid males and females. There was a slight but nonsignificant tendency for females from nests with diploid males to have higher homozygosity (average proportion of genome in ROH blocks >10 kbp for females from nests with diploid males = 34.4%, nests without diploid males = 31.6%; S2B Fig).

We then used the genomic distribution of ROH to identify candidate regions for CSD. We divided the ROH into genomic segments of uneven size across all individuals. Using a linear model, we associated the presence or absence of ROH segments across the genome with the sex of the diploid individuals. This analysis identified a single 2 kbp candidate region on chromosome 1 with coordinates 14,404,983–14,406,957 in which the presence of ROH is highly associated with sex (Fig 2A; $p < 10^{-16}$ after adjustment for multiple testing). All diploid males, and none of the females, are completely homozygous across this region. This is the only region of the genome where strong differentiation is observed in our dataset and is therefore likely to contain the CSD locus.

Genomic regions responsible for CSD are expected to evolve under frequency-dependent selection, which results in the presence of a large number of divergent alleles to avoid the occurrence of homozygosity in diploids [15]. This process leads to high levels of nucleotide diversity at CSD loci. We found that the candidate CSD region coincides with a local maximum in nucleotide diversity estimated in 10 kbp windows across the genome (Fig 2B). Nucleotide diversity in the region is 0.017, which is an order of magnitude higher than the chromosome average (mean$_{CHR}$ = 0.0014, stdev$_{CHROM}$ = 0.00087). It is the highest peak on chromosome 1, the fourth highest peak genome-wide, and in the top 99.86% of 10 kbp bins genome-wide. This supports the action of frequency-dependent selection in the region.

The candidate CSD region overlaps a single uncharacterized lncRNA in the genome annotation (Fig 2C–2E). The identity and order of genes flanking this lncRNA are identical to those flanking *ANTSR*, the lncRNA implicated in sex determination in the ant *L. humile* [10]. These include COPA, CRELD2, miR-315, and THUMPD3 (Fig 2E). This indicates that the CSD locus identified here overlaps the *ANTSR* ortholog. *ANTSR* has 2 exons annotated in the *O. bicornis* genome assembly and the 2 kbp CSD locus overlaps 272 bp of the intron, 354 bp of exon 2, and 1,348 bp of downstream sequence.

## High genetic diversity in CSD concentrated in specific subregions

In total, we identified 197 SNPs and 26 indels in the CSD locus. We identified 188 distinct haplotypes in the full sample of 254. A tree showing the relationships between haplotypes reveals high diversity across several clades (Fig 3A). We

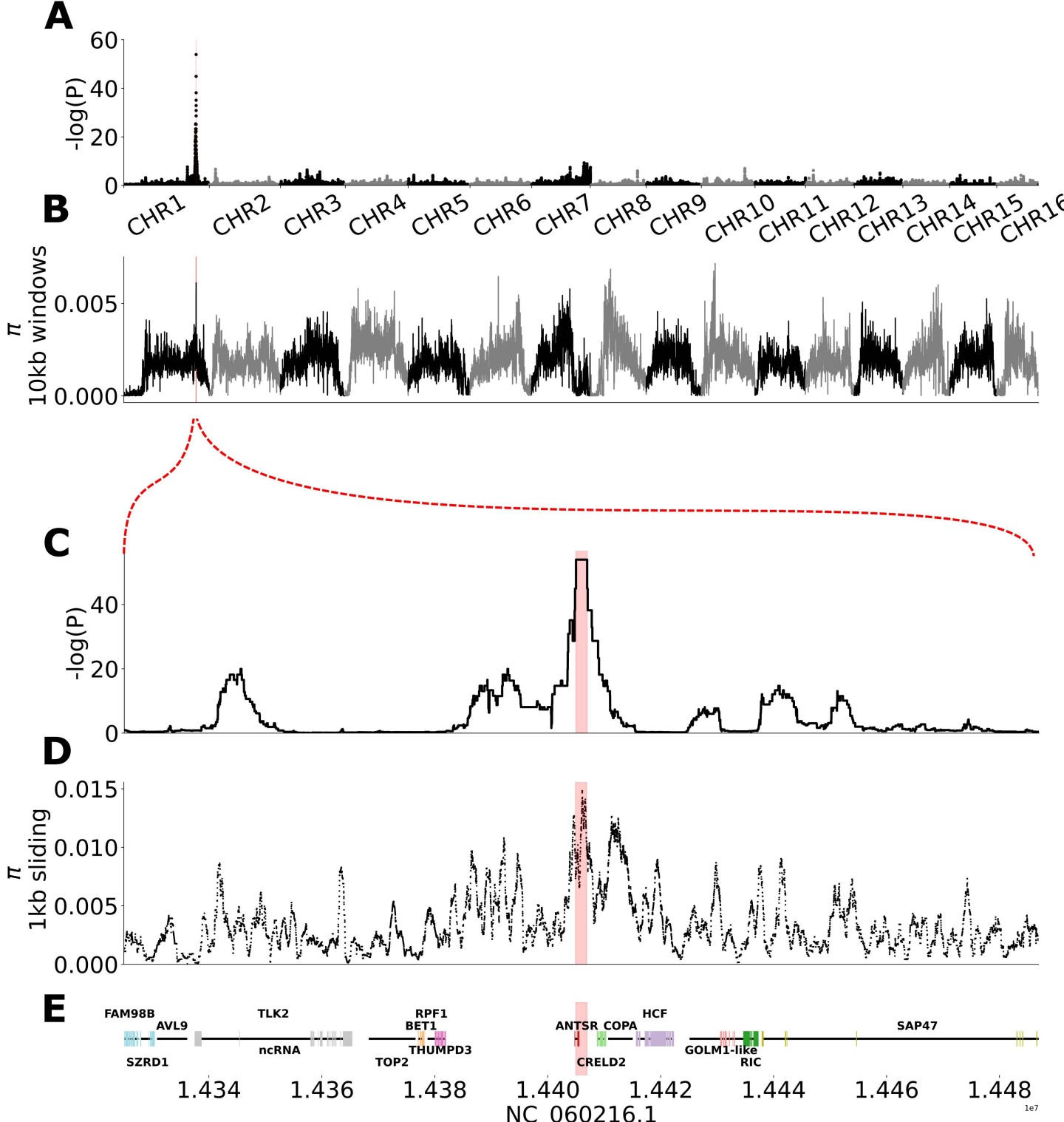

**Fig 2. Mapping the CSD locus. A**. Association of presence/absence of runs of homozygosity (ROH) with sex in diploid samples using a linear model (sex~ROH; $n = 18$ diploid males, $n = 74$ females; correction for genome-wide significance using Benjamin–Hochberg procedure). **B.** Nucleotide diversity in 10 kbp windows across the genome. **C**. Close-up of association results on chromosome 1. The red dotted lines between panels B and C indicate the

location on Panels A and B. **D.** Nucleotide diversity across the candidate region estimated in 1 kbp sliding windows. **E.** Genomic features in and around the candidate region. The red vertical bar indicates the best-scoring ROH segment across all subplots. Data from panels **A–E** can be found in files Data1–Data3 within https://doi.org/10.5281/zenodo.17424723 and S5 Table.

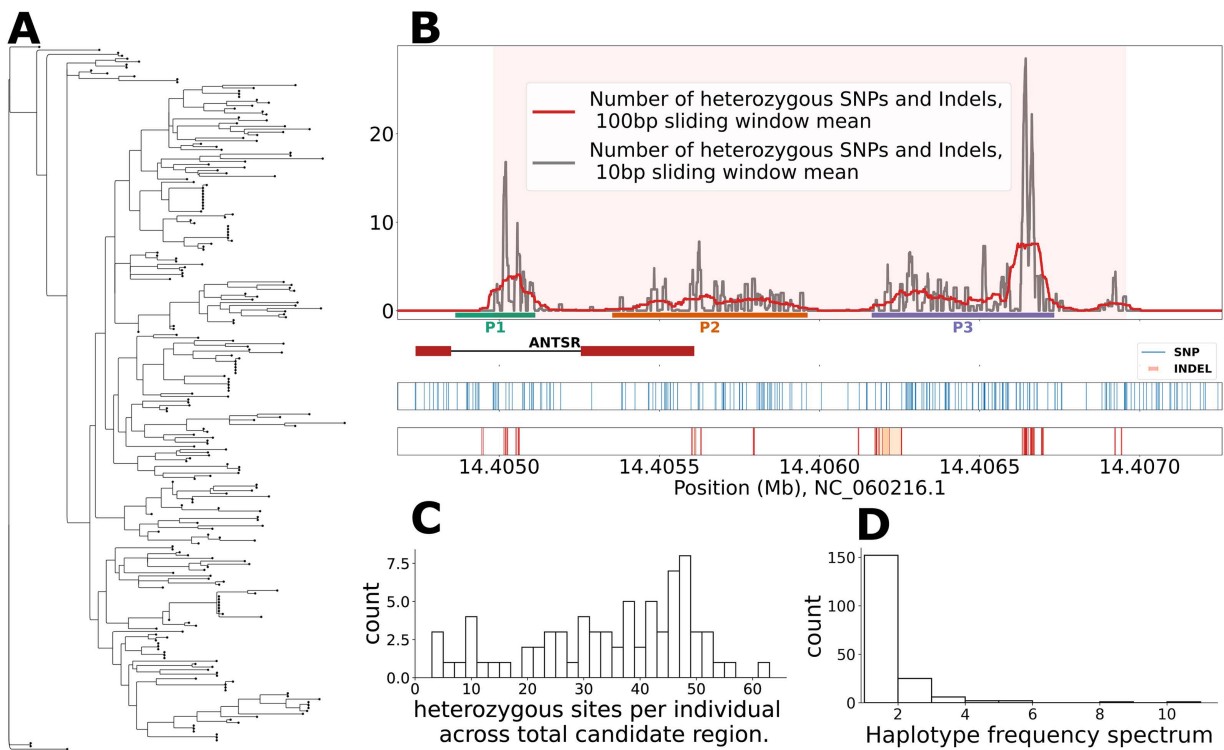

**Fig 3. Haplotypes across the 2 kbp CSD locus. A.** Maximum-likelihood tree of all unique haplotypes across the mapping population (74 females, 70 haploid males, 18 diploid males, 254 total haplotypes). Circles at branch tips indicate the number of observations of each haplotype. **B.** Count of differences between the two haplotypes carried by females across the candidate region. The top panel shows differences across all females, with the candidate region highlighted in light red. Three regions with larger differences within the region are labeled P1, P2, and P3, highlighted with blue, green, and orange bars at the bottom of the panel, respectively. The middle panel shows how *ANTSR* is located within the candidate region. The bottom two panels show a rug plot of heterozygous SNPs across the region in blue, with indel starting positions shown as red bars, and their extent in orange. **C.** Histogram of the total number of heterozygous sites across the candidate region per individual. **D.** Histogram of the haplotype frequency spectrum showing the number of times each haplotype is observed in the population samples. Data for panels **A–D** can be found in files Data4–Data7 within https://doi.org/10.5281/zenodo.17424723.

examined the distribution of heterozygous sites in females across the CSD region to attempt to identify critical regions where heterozygosity is required to activate female development (Fig 3B). Heterozygosity in the *ANTSR* exons within the CSD locus does not seem to be required to trigger female development as 7/74 females do not have any heterozygous sites in the overlap between the candidate region and exonic regions. The distribution of heterozygous sites in females is variable across the CSD locus. We define three regions where heterozygosity is particularly elevated (S5 Table): in an intron (P1), overlapping the final exon and downstream noncoding region (P2), and a downstream noncoding region (P3). These regions are particularly notable for a high number of indel polymorphisms. These candidate functional regions could potentially affect female development by influencing transcription levels of *ANTSR* (S6 Table).

We found that the two haplotypes present in females at this locus differ on average at 24.6 SNPs and 9.5 indels, but there is a large variation in this number (Fig 3C). We examined the importance of heterozygosity in the three regions we

defined (S3 Fig). We identified 11 females that were homozygous across one of these regions and one that was invariable across two regions (S6 Table). The minimum number of differences observed between the two haplotypes in a female in our dataset is 3 SNPs, which is found in two individuals. These observations suggest that none of these regions is critical for determining sex on its own, and that heterozygosity at one or more of a large number of critical sites across these regions is required for triggering female development, indicative of genetic heterogeneity.

The majority of haplotypes ($n = 152$) are seen only once in the sample, with 36 haplotypes observed multiple times (Fig 3D). The frequency of diploid male production in a randomly mating population is expected to be approximately $1/k$, where $k$ is the number of sex alleles [4]. In our populations, this proportion is ~0.53% (S4 Table), which is the same as the theoretical expectation of 0.53% assuming that there are 188 haplotypes at the CSD locus, although it is likely that additional haplotypes exist in the population that are not observed in our sample.

## Elevated nucleotide diversity implicates *ANTSR* in sex-determination in other bee and ant species

We identified the location of the *ANTSR* ortholog in the genomes of nine other bee species (*Apis mellifera*, *Anthophora quadrimaculata*, *Anthophora retusa*, *Anthophora plagiata*, *Bombus monticola*, *Bombus lapponicus*, *Bombus balteatus*, *Lasioglossum albipes*, and *Lasioglossum pauxillum*) and three ant species (*Formica polyctena*, *Formica aquilonia,* and *Solenopsis invicta*; Fig 4A and 4B; S7 Table). In each case, we could identify a candidate *ANTSR* ortholog including the same set of flanking genes as in *O. bicornis*, indicating a conserved genomic neighborhood.

We assayed variation in heterozygosity in 10-kbp windows across the genomes of these species. There is no elevated variation in the orthologous region of *Apis mellifera*, in which sex determination is known to be controlled at a locus unrelated to *ANTSR* [8,16]. In all other species of bees and ants that we investigated, we find markedly elevated genetic variation in the vicinity of the *ANTSR* ortholog (S8 Table). In *A. quadrimaculata, A. retusa, B. monticola, B. lapponicus, L. pauxillum,* and *L. albipes,* we observed a peak of elevated genetic variation in the 10 kbp window containing *ANTSR* (genetic variation in the top 0.3% of windows genome wide). In *A. plagiata*, *B. balteatus*, *S. invicta, F. polyctena,* and *F. aquilonia,* such a peak was found between 0.5 and 45 kbp (S8 Table), away from *ANTSR*.

These findings are consistent with the action of frequency-dependent selection in the vicinity of *ANTSR* in all of the bee and ant species we investigated, with the exception of *Apis mellifera*. This lncRNA gene is therefore likely to be the sex-determining locus in all of these species in addition to *Osmia bicornis*. Together, our findings suggest that *ANTSR* is the predominant sex-determining gene in bees and ants, and likely had this role in their common ancestor ~153 million years ago [13,14].

## Discussion

Single-locus CSD is found widely in Hymenoptera and is likely to be the ancestral mode of sex determination [4,5,7,15]. CSD loci have been characterized in one ant (*L. humile*) and one bee (*A. mellifera*) species [8,10], mapping to two non-orthologous loci. Here we identify the CSD locus in the solitary bee *O. bicornis* and analyze haplotype variation at this locus. Our main results are: (1) a single CSD locus in *O. bicornis* maps to an ortholog of the *ANTSR* lncRNA gene, which is known to determine sex in the ant *L. humile*, (2) a large number of highly divergent haplotypes are observed at the *O. bicornis* CSD locus, with elevated diversity in specific sub-regions, (3) elevated levels of nucleotide diversity are found in or near orthologs of the *O. bicornis* CSD locus in three other bee genera: *Anthophora*, *Bombus,* and *Lasioglossum*, and two ant genera: *Formica* and *Solenopsis*. Together, the results indicate that *ANTSR* is likely to be the ancestral sex-determination gene of bees and ants and could be the predominant mechanism of sex determination in Hymenoptera.

Downstream sex determination in insects acts through expression of alternative splice variants of the gene transformer (*tra*) or a related homolog, leading to male or female development [19]. In most Hymenoptera, splicing of *tra* is determined by zygosity at a CSD locus [4,7]. Two possible modes of action of CSD loci were originally suggested: (1) formation of a heteropolymer consisting of the expressed product of two alleles at CSD, and (2) production of mRNA defective in one or more cistrons, with different

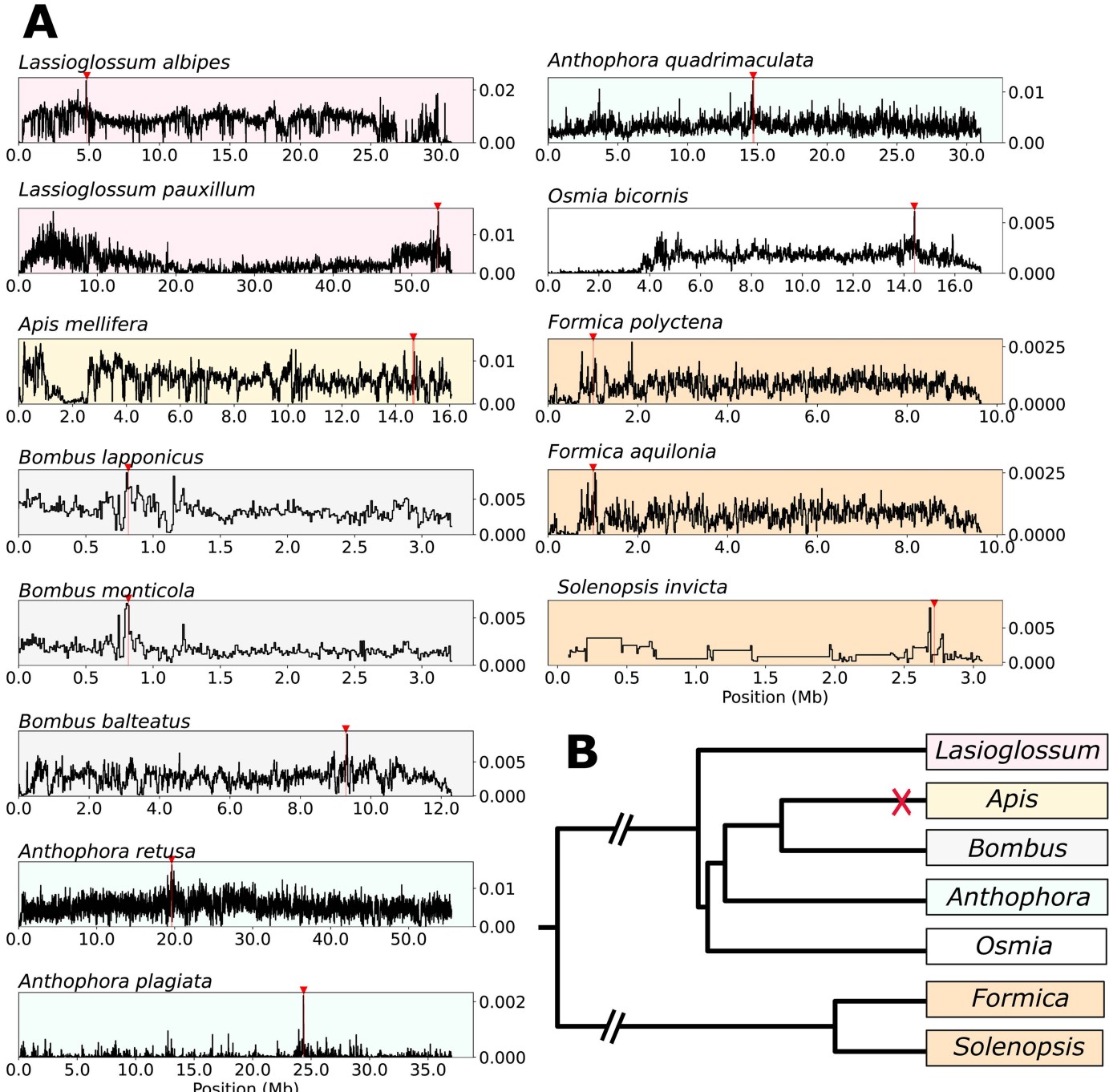

**Fig 4. Nucleotide diversity around *ANTSR* in other bee and ant species. A.** Variation in nucleotide diversity (π) is shown in 10 kbp windows across the contig containing *ANTSR* in each of the following species: *Lassioglossum albipes* and *Lassioglossum pauxillum* (both shaded red) *Apis mellifera* (shaded yellow)*, Bombus lapponicus*, *Bombus monticola*, *Bombus balteatus* (all *Bombus* shaded gray), *Anthophora retusa*, *Anthophora plagiata*, *Anthophora quadrimaculata* (all *Anthophora* shaded green), *Osmia bicornis* (shaded white), and three ant species (*Formica polyctena*, *Formica aquilonia*, and *Solenopsis invicta,* all shaded orange*).* The vertical red line and red arrow above the plots indicate the location of the *ANTSR* ortholog for each species. **B.** Phylogenetic tree indicating the relationship between the investigated genera. The tree is based on the distance matrix from BeeTree [25], and Romiguier and colleagues [53]. The data underlying the nucleotide diversity plots can be found in files Data2 and Data8–Data19 within https://doi.org/10.5281/zenodo.17424723.

alleles containing different defective cistrons [4]. The first mechanism has been demonstrated in honeybee, where formation of a heterodimer of the Csd protein is required to generate alternative splice variants of *fem* (ortholog of *tra*) [9].

The CSD locus identified here overlaps the lncRNA *ANTSR*, which was identified as the sex-determining locus in the ant *L. humile* [10]. This locus has also been implicated in sex determination by a genetic mapping study in the clonal raider ant, *Ooceraea biroi* [20], which also identified orthology to a CSD quantitative trait locus previously found in the ant *Vollenhovia emeryi* [21]. Its mechanism of action is not fully understood, but it is not consistent with either of the two originally proposed mechanisms above. In *L. humile, ANTSR* has four exons and encodes a 793-bp transcript. The CSD locus was identified as a 5 kbp region overlapping two of the *ANTSR* exons. The exonic sequences are relatively low in variation and are identical in some haplotypes, indicating that heterozygosity in exons does not generate the sex determination signal. Variability in CSD is mainly restricted to downstream flanking and intronic sequence of *ANTSR* and heterozygosity in these regions appears to regulate expression of *ANTSR*, which in turn affects splicing of *tra* [10]. Our findings in *O. bicornis* are consistent with those in *L. humile*. The *ANTSR* ortholog in *O. bicornis* has only two exons, which also completely lack variability in some females, with variability in CSD also focused on intronic and downstream flanking sequence. It therefore seems likely that heterozygosity at the CSD locus increases expression of *ANTSR* in both *O. bicornis* and *L. humile* [10].

We identified 188 distinct haplotypes in the 2 kbp CSD identified in *O. bicornis*. The high haplotype diversity found in *O. bicornis* makes it possible to more precisely define hypervariable regions of CSD that might be important in its function. We identified three sub-regions with particularly elevated density of SNPs and indels, which overlap an intron and downstream noncoding sequence. However, we identified females that lack variability in one or two of these regions, indicating that heterozygosity at a small number of many possible critical sites distributed across these three sub-regions is sufficient to trigger female development. A plausible mechanism that has been proposed is transvection [10], where the regulatory region of a gene can influence expression in *trans*. This mechanism is likely to be common in insects in which homologous chromosomes remain paired in somatic cells [22] and is responsible for sex-biased expression of an X-linked gene in *Drosophila* [23].

Only seven haplotypes were identified at the 5 kbp CSD locus identified in *L. humile* [10], likely because the population of *L. humile* used to map CSD was highly inbred, which leads to a high frequency of diploid male production. The production of diploid males represents a significant cost to CSD, particularly in inbred populations. This cost could be partly ameliorated by more efficient purging of deleterious mutations in haplodiploids [24]. We estimate that diploid male production occurs at a rate of ~0.5% of matings in our *O. bicornis* population. This low level of diploid male production is consistent with high haplotype diversity, but nonetheless represents a substantial genetic load associated with CSD.

We have identified elevated genetic variation close to *ANTSR* in 13 species belonging to six bee and ant genera: *Anthophora, Bombus, Lasioglossum, Osmia, Formica,* and *Solenopsis*. This indicates that the CSD locus is associated with *ANTSR* in these genera. The observation that the peak does not always correspond exactly to *ANTSR* (e.g., in *A. quadrimaculata, B. balteatus, F. polyctena, F. aquilonia,* and *S. invicta*), supports the view that CSD is a non-coding regulatory element that regulates *ANTSR*, which has shifted location in evolution.

Our results are consistent with previous research demonstrating that the CSD locus in *Apis mellifera* (*Csd*) is a recent innovation, which arose from a duplication of the *feminizer* gene [16]. As *Bombus* is a close relative of *Apis* (split ~54 million years ago) [14,25] but likely uses *ANTSR* for sex-determination, it is likely that it represents the ancestral state. Taken together, our results suggest that *ANTSR* is the ancestral sex determination locus of Apoidea (bees and sphecoid wasps) and Formicidea (ants). These superfamilies encompass around 50,000 species, or around one-third of Hymenoptera. It is also plausible that *ANTSR* could determine sex much more widely in Hymenoptera.

## Materials and methods

### Sample collection

*O. bicornis* is a univoltine, protandrous, sexually size-dimorphic, polylectic, and monandrous spring bee. Each individual hibernates as an adult inside a cocoon in the maternal nest. Females construct typically mixed nests of line type

containing both sexes. A series of larger cells for daughters is constructed at the back of the tube and a series of smaller cells for sons at the entrance [26] (Fig 1C). Body size of an individual is limited by the provisions in the brood cell [27]. Males developing in a female-dedicated brood cell with more provisions reach a female-like body mass. This situation occurs when the eggs are diploid male (due to homozygosity at the CSD locus) or due to fertilization errors, which result in a haploid egg. The rate of fertilization errors has previously been estimated to be about 6% in female-dedicated eggs [28].

Diploid males occur rarely and are detectable as adults with phenotypic male characteristics in female-dedicated brood cells. We analyzed the contents of 683 *Osmia bicornis* nests from the botanical gardens Halle (Saale), Germany and from various locations in the vicinity of Uppsala, Sweden during the period 2021–2023 to identify potential diploid males. The nests were contained in Japanese knotweed in Halle (internal diameter 8–10 mm, length 133.4 ± 13.2 mm) or cardboard tubes in Uppsala (internal diameter 7 mm, length 150 mm). They were analyzed during the winter, when adult bees are dormant inside cocoons.

Males and females were distinguished by morphological examination of the contents of each cocoon. Bees with black hairs on the clypeus were considered female, whereas the presence of long white hairs indicated a male. Potential diploid males were identified as morphological males that satisfied these criteria: (1) presence in a female-dedicated brood cell or (2) mass greater than 0.1 g, or both. Ploidy of all potential diploid males was determined by subsequent genome sequencing. We also collected females from nests containing potential diploid males and from random nests for sequencing.

## Library preparation and sequencing

Samples were stored in 95% ethanol at −20°C prior to DNA extraction. They were dissected to extract wing muscle from the thorax. DNA was extracted using the Qiagen DNeasy Blood and Tissue Kit. We performed sequencing library preparation using the Illumina Nextera Flex kit. Sequencing was performed using an Illumina NovaSeq 6000 (output per sample shown in S1 Table).

## Read mapping, variant calling, and filtering

Reads from all samples were mapped to the *Osmia bicornis* genome assembly iOsmBic2.1 using bwa v0.7.17 [29]. The resulting files were sorted and indexed using samtools 1.19 [30] before using Picard v2.25.0 [31] to mark duplicates and add read groups. Subsequently, GATK HaplotypeCaller v4.2.0.0 [32] was used to call variants. We considered all samples as diploids for variant calling. The software vcftools v0.1.16 [33] was used to filter the dataset, retaining only biallelic SNPs, MQ > 20, AF ≥ 0.01, as well as missingness <0.1. We also filtered markers for substandard or excess depth (1,000 > DP > 3,000) and quality by depth (QD > 20).

We noted that in certain regions of the genome, genotype calls had a large excess of heterozygosity, read mapping depth was significantly higher than average and quality-by-depth was significantly lower (S1A and S1B Fig). This observation is consistent with the presence of long stretches of highly repetitive heterochromatin in the genome assembly, in which genetic variants cannot be reliably assayed. This has been reported in a wide range of other genomes [34]. We identified regions with suspected heterochromatin based on elevated sequencing depth, reduced quality-by-depth and excess heterozygosity. To define these regions, we ran vcf-tools with the *--hardy* option, plotting $-\log(P_{het\_excess})$. We manually excluded regions of excess heterozygosity that were observed at the distal ends of each of the 16 chromosomes (S1A and S1B Fig; S9 Table). These blocks were excluded for the determination of ploidy and all further genome-wide summary statistics. Additionally, we filtered all markers identified as heterozygous in at least two haploid individuals from the dataset for mapping in the diploid individuals. The regions comprise 23.8% of the total assembled chromosomes. We also excluded 3 samples due to low average sequencing depth (<2 million reads).

## Identification of ploidy

We used the --het option in vcftools [33] to calculate the inbreeding coefficient *F*, the ratio of observed and expected homozygotes per individual. This statistic revealed the ploidy of all samples, and determined whether potential diploid

males were haploid or diploid. The male samples clustered into clear groups, with those with ($F > 0.95$) considered to be haploid males due to lack of heterozygous sites (S1C and S1D Fig).

### Estimation of the frequency of matched matings

We used the number and distribution of confirmed diploid males among nests to estimate the frequency of matched matings in the populations. A matched mating is defined as one where a female mates with a male that shares one of its CSD haplotypes, which results in a probability of 0.5 that each fertilized egg will become a diploid male. We estimated the proportion of matched matings by dividing the number of nests containing diploid males with the total number of nests containing females, with a correction based on the likelihood of detecting a matched mating based on independent segregation of alleles at a single CSD locus. This likelihood is defined as $d = 1 - 0.5^n$, where $n$ is the number of diploid offspring present in a nest. We used this estimate to define the effective number of nests, by summing the values of $d$ across all nests analyzed according to the number of diploid offspring per nest (S4 Table). This was performed using data collected in Halle over two consecutive years, and from 1 year in Uppsala.

### Differences in mass

Using the ploidy confirmed by sequencing and the recorded sex, we tested for mass differences between cocoons containing females, haploid males, and diploid males. To compare diploid males to their female nestmates, we used a subset of 15 diploid males with multiple sequenced sisters ($n = 37$). We used statsmodels in python to make a linear mixed-effect model to test for an association between mass in grams and sex encoded as 0 and 1, with NestID as a random effect. We tested for differences between the mean mass of all confirmed diploid males ($n = 18$), females ($n = 1,745$), and haploid males, for which data was available ($n = 1,659$), using two-sided unequal variance $t$ tests.

### SNP density and nucleotide diversity

Nucleotide diversity (π) and SNP density were calculated in 10 kbp windows basis using vcftools [33] with the --window-pi option. For fine mapping nucleotide diversity in the identified putative CSD region, π was calculated per site using vcftools and the --site-pi option, and subsequently evaluated in a 1-kbp window rolling mean using numpy v1.26.0 [35].

### Identification of runs of homozygosity associated with sex

ROH were calculated for each individual using plink v1.9 [36] with the *---homozyg group --homozyg-window-snp 50 --homozyg-window-het 3 --homozyg-window-missing 5 --homozyg-window-threshold 0.05 --homozyg-snp 10 --homozyg-kb 5 --allow-extra*-chr and otherwise default options. Then, using pandas (2.2.2; [37]) and numpy (1.26.0; [35]) in python 3.10, we segmented the resulting ROH into a binary square matrix across the individuals and the genome, with the borders across the latter being defined by the beginning or end of a ROH within any of the investigated individuals. Subsequently, we tested the association between presence or absence of a ROH segment with sex in diploid males and females with a linear model, using statsmodels (0.14.2; [38]) and controlled for the rate of false positive testing using the Benjamini–Hochberg [39] procedure implemented in statsmodels with $\alpha = 0.05$.

### Haplotype analysis

In order to investigate the haplotype in the candidate region, we extracted the highest-scoring ROH-block (1974 bp), phased it using beagle v5.5 [40], and generated a sequence in fasta format for each haplotype. These were subsequently aligned using MAFFT (v7.520; [41]) and used to generate a tree using RAxML v8.2.12 [42], which was plotted using FigTree v1.4.4 [43].

## Comparison with other species

We used blast hits identified with the blastn tool of BLAST+ v. 2.16.0 [44] to locate the ortholog of *ANTSR* in several other bee species for which genetic variation data were available. We created a query list using the *ANTSR* sequence of *L. humile* and its orthologous sequences in *A. mellifera* (LOC100576403) and *Bombus affinis* (LOC125386106) reported by Pan and colleagues [10]. We also included the *O. bicornis* lncRNA sequence studied here. We blasted the four query sequences against two *Bombus*, three *Anthophora,* two *Lasioglossum, and three* ant genome assemblies. These were *Bombus sylvicola* (BioProject PRJNA646847) [45]*, Bombus balteatus* (PRJNA704506) [46]*, Anthophora retusa, Anthophora quadrimaculata*, and *Anthophora plagiata* (PRJNA1255204) [47]. *Lasioglossum pauxillum* (PRJNA13330011), *Lasioglossum albipes (*PRJNA413432) [48]*, Formica polyctena, Formica aquilonia* (PRJEB51899), [49] and *Solenopsis invicta* (PRJNA685290) [50]. The region with the highest query cover and identity was identified for each species.

We then examined the annotations of protein-coding and non-coding RNA genes in these regions for each target genome to look for the presence of an annotated lncRNA gene and conservation of the flanking genes. Functional annotations were not available for all of the species. We therefore performed functional annotation for these genome assemblies based on orthology using eggNOG-mapper v.2 [51]. This was performed with eggNOG-mapper default values against the eggNOG 5 database.

We identified the most likely position of *ANTSR* in each genome based on the annotation of a lncRNA in the vicinity of the best BLAST hit. A candidate gene was present in the annotation of some species, but not available for *A. quadrimaculata*, *L. pauxillum, L. albipes, F. polyctena, F. aquilonia,* and *S. invicta*. For these species, the best BLAST hit was used as the putative location. For *A. mellifera*, we used the coordinates previously reported by Pan and colleagues [10]. We analyzed genetic diversity in sequenced populations of each of these species. We used 65 and 49 individuals for *Bombus lapponicus* and *Bombus monticola*, respectively, mapped to the reference genome of the closely related species *B. sylvicola* [45]. For *B. balteatus,* we used 299 bees mapped to the *B. balteatus* genome [46]. We used 57, 43, and 27 individuals for *A. retusa, A. quadrimaculata*, and *A. plagiata* [47]. We used 55 *Apis mellifera* individuals (BioProject PRJNA357367) [52]. We used 71 individuals for *L. pauxillum,* 160 for *L. albipes,* 10 individuals for each of the two *Formica* species, and 368 for *S. invicta*. A summary of datasets is presented in S7 Table. We used vcftools v.0.1.16 [33] to calculate genetic diversity based on the pairwise differences (π) in windows of 10 kbp across the genome using vcftools --window-pi. We generated a phylogeny of these species using BeeTree [25], plotted using FigTree 1.4.4 [43].

## Supporting information

**S1 Fig. Presence of heterochromatin at the distal ends of chromosomes and its effect on the inbreeding coefficient, *F*. A:** Manhattan plot of *p*-values for excess heterozygosity per marker across all individuals and the 16 largest chromosomes. **B:** Top panel: Manhattan plot of *p*-values for excess heterozygosity per marker across all individuals and Chromosome 1. Bottom panel: read depth (y-axis) and Quality-by-depth (color) across chromosome 1. **C:** Effect of filtering heterochromatin regions on *F*. *F* derived from the unfiltered genome on the x-axis, *F* derived from the filtered genome on the y-axis. Females colored in red, haploid males initially suspected to be diploid in blue, correctly identified diploid males in gray. **D:** Strip plot of *F* after filtering. Females colored in red, haploid males initially suspected to be diploid in blue, correctly identified diploid males in gray. Data underlying panels A–D can be found in files Data20–Data22 within https://doi.org/10.5281/zenodo.17424723.
(EPS)

**S2 Fig. Distribution of weights and ROH levels in all samples. A:** Kernel density estimate plot and strip plot of weight distribution of all females and haploid males from nests collected in Uppsala. Red line and dots: Females. Yellow line and

dots: haploid males. Blue line and dots: diploid males. **B:** Kernel density and strip plot of the percentage of the genome covered with ROH bigger than 10 kbp using the full dataset of 162 sequenced samples. Solid blue line: diploid males. Solid red line: all females. Dashed red line: females from nests containing a diploid male. Dotted red line: females collected from nests without diploid males. Data underlying panel A can be found in S3 Table. Data underlying Panel B can be found in files Data24 within https://doi.org/10.5281/zenodo.17424723.
(EPS)

**S3 Fig. Number of heterozygous sites across the candidate region: from left to right: total candidate region, region P1, P2, and P3.** See Fig 3 and S5 Table for coordinates. Data underlying this figure can be found in files Data5 within https://doi.org/10.5281/zenodo.17424723.
(EPS)

**S1 Table. Summary of all samples sequenced.**
(XLSX)

**S2 Table. Summary of *Osmia bicornis* cocoons containing potential diploid males collected in Halle, Germany over 2 consecutive years.**
(XLSX)

**S3 Table. Summary of *Osmia bicornis* cocoons containing potential diploid males collected in Uppsala.**
(XLSX)

**S4 Table. Estimation of frequency of matched mating.**
(XLSX)

**S5 Table. Coordinates of elements within the candidate region of *Osmia bicornis*.**
(XLSX)

**S6 Table. Number of heterozygous sites (SNP and INDELS) per female individual across peaks in candidate-region and total candidate region.**
(XLSX)

**S7 Table. Information on Hymenoptera species used in comparative analysis.**
(XLSX)

**S8 Table. Nucleotide diversity in the Orthologous region to *ANTSR*, by species.**
(XLSX)

**S9 Table. Estimated heterochromatin regions on each *Osmia bicornis* chromosome.**
(XLSX)

## Acknowledgments

We thank Adriana M Cintrón Santiago and Tuuli Larva for help with DNA extractions. We thank Hanna Sigeman and Miguel Carneiro for helpful comments on the manuscript. The authors acknowledge support from the National Genomics Infrastructure in Stockholm funded by Science for Life Laboratory, the Knut and Alice Wallenberg Foundation and the Swedish Research Council, and NAISS/Uppsala Multidisciplinary Center for Advanced Computational Science for assistance with massively parallel sequencing and access to the UPPMAX computational infrastructure. The computational analysis was enabled by resources provided by the National Academic Infrastructure for Supercomputing in Sweden (NAISS), partially funded by the Swedish Research Council through grant agreement no. 2022–06725.

## Author contributions

**Conceptualization:** Björn Cederberg, Robert J. Paxton, Karsten Seidelmann, Matthew T. Webster.

**Data curation:** Tilman Rönneburg, Demetris Taliadoros, Turid Everitt, Panagiotis Theodorou, Karsten Seidelmann, Matthew T. Webster.

**Formal analysis:** Tilman Rönneburg, Demetris Taliadoros, Panagiotis Theodorou, Karsten Seidelmann, Matthew T. Webster.

**Funding acquisition:** Matthew T. Webster.

**Investigation:** Tilman Rönneburg, Anna Olsson, Sara Magnusson, Linn Huser, Muhammad Nafiz Ikhwan Bin Nor Fuad, Björn Cederberg, Karsten Seidelmann, Matthew T. Webster.

**Methodology:** Tilman Rönneburg, Karsten Seidelmann, Matthew T. Webster.

**Supervision:** Matthew T. Webster.

**Visualization:** Giselle C. Martín-Hernández.

**Writing – original draft:** Tilman Rönneburg, Matthew T. Webster.

**Writing – review & editing:** Tilman Rönneburg, Demetris Taliadoros, Björn Cederberg, Robert J. Paxton, Karsten Seidelmann, Matthew T. Webster.

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
