## [Editor Report · Decision Letter 0]

10 Jul 2025

Dear Dr Webster,

Thank you for submitting your manuscript entitled "Genetic mapping in the red mason bee Osmia bicornis implicates ANTSR as an ancient sex-determining locus in bees and ants" for consideration as a Research Article by PLOS Biology.

Your manuscript has now been evaluated by the PLOS Biology editorial staff, as well as by an academic editor with relevant expertise, and I'm writing to let you know that we would like to send your submission out for external peer review.

IMPORTANT: We think that it would be best to consider your paper as a Short Report. It's already nice and concise, so no reformatting is needed at this time, but please change the article type to "Short Reports" when you upload the additional metadata (see next paragraph).

Once your full submission is complete, your paper will undergo a series of checks in preparation for peer review. After your manuscript has passed the checks it will be sent out for review. To provide the metadata for your submission, please Login to Editorial Manager (https://www.editorialmanager.com/pbiology) within two working days, i.e. by Jul 14 2025 11:59PM.

Kind regards,

Roli Roberts

Roland Roberts, PhD

Senior Editor

PLOS Biology

rroberts@plos.org

---

## [Decision Letter · Decision Letter 1]

22 Aug 2025

Dear Dr Webster,

Thank you for your patience while your manuscript "Genetic mapping in the red mason bee Osmia bicornis implicates ANTSR as an ancient sex-determining locus in bees and ants" was peer-reviewed at PLOS Biology. It has now been evaluated by the PLOS Biology editors, an Academic Editor with relevant expertise, and by two independent reviewers.

Based on the reviews, we are likely to accept this manuscript for publication, provided you satisfactorily address the remaining points raised by the reviewers and the following data and other policy-related requests.

IMPORTANT - please attend to the following:

a) Please remove the Linnaean name of the study species from the Title, as it appears clearly in the Abstract: "Genetic mapping in the red mason bee implicates ANTSR as an ancient sex-determining locus in bees and ants"

b) Please address the reviewers' concerns. We discussed the more onerous requests from reviewer #2 with the Academic Editor; we think that for this Short Report format, the experimental (functional) work is beyond the scope of the study. However, it should be possible to include more genomes in the comparative analysis, and this should strengthen your findings - please let me know if you need additional time to do this. You should also makesure that you adjust any claims to be commensurate with the evidence presented.

c) Please address my Data Policy requests below; specifically, we need you to supply the numerical values underlying Figs 2ABCD, 3ABCD, 4, S1ABCD, S2AB, S3, either as a supplementary data file or as a permanent DOI’d deposition.

d) Please cite the location of the data clearly in all relevant main and supplementary Figure legends, e.g. “The data underlying this Figure can be found in S1 Data” or “The data underlying this Figure can be found in https://zenodo.org/records/XXXXXXXX

e) Please make any custom code available, either as a supplementary file or as part of your data deposition.

We expect to receive your revised manuscript within two weeks.

*Published Peer Review History*

*Press*

Sincerely,

Roland

Roland Roberts, PhD

Senior Editor

rroberts@plos.org

PLOS Biology

DATA POLICY:

Regardless of the method selected, please ensure that you provide the individual numerical values that underlie the summary data displayed in the following figure panels as they are essential for readers to assess your analysis and to reproduce it: Figs 2ABCD, 3ABCD, 4, S1ABCD, S2AB, S3. NOTE: the numerical data provided should include all replicates AND the way in which the plotted mean and errors were derived (it should not present only the mean/average values).

CODE POLICY

DATA NOT SHOWN?

REVIEWERS' COMMENTS

Reviewer #1:

This manuscript maps the locus responsible for complementary sex determination in the red mason bee, by screening the genomes of diploid females and rare diploid males for regions where homozygosity is associated with sex. The authors identify a 2 kbp locus that contains ANTSR, a sex-determining gene previously identified in ants, and uncover small tracts of unusually high diversity overlapping and immediately adjacent to ANTSR. They further show peaks of genetic diversity at ANTSR in six other bee species, and the absence of a diversity peak in honeybee, which uses a different gene for sex determination. This strongly suggests that ANTSR is the ancestral sex determination gene for bees and ants. A fine-scale analysis of which tracts within the ANTSR region do or don't contain heterozygous SNPs in each female demonstrates that no single position within the ANTSR region is critical for female development, but that a minimum of three heterozygous SNPs within three high-diversity tracts seem to be required.

This study substantially pushes back the minimum age of this sex-determining mechanism, and greatly expands the number of species in which we would expect to find it operating. It also provides clues toward the mechanism by which ANTSR determines sex, and the functionally important regions within in and around the gene. The study is technically sound and well presented in the text and figures, and I could find no suggestions for further improvement. I congratulate the authors on an excellent contribution.

Reviewer #2:

The authors have used state-of-the-art genomics analysis and a clever experimental design to identify a gene implicated in the regulation of haplodiploidy. Haplodiploidy is one of the major forms of sex determination, made famous by the fact that it is the form of sex determination found in the ants, bees, and termites, which contain the majority of the world's species that exhibit eusociality and form complex societies. Understanding the molecular basis of complementary sex determination (CSD), the genetic mechanism underlying haplodiploidy, is thus of major interest in biology and of interest to many types of biologists, from hard-core geneticists to those interested in the mechanisms of social evolution. In 2003 Martin Beye and colleagues published a landmark paper in Cell identifying the CSD gene of the honeybee, named csd. Since then Beye and other labs have established functional connections between csd and some of the "traditional" sex determination machinery in Drosophila. One other CSD gene has been identified, ANTSR, in an ant species. Because both ants and honey bees are highly derived species, the origins of CSD genes remains unknown.

This is the topic tackled by Ronneburg et al. in this manuscript. The authors sequenced the genome of the Red Mason Bee, a Megachilid much more basal than the other two species above. To scan the genome efficiently for a putative CSD gene, they cleverly used diploid males, comparing genome sequences and allelic frequencies with diploid females and normal haploid males. Under haplodiploidy diploid males are developmentally inviable, but using a robust "needle in a haystack" strategy they identified 6 individuals in 672 nests and got the sequence information needed to pinpoint ANTSR again. Because of the phylogenetic position of the Red Mason Bee, the authors assert that ANTSR is the ancestral CSD gene in Hymenoptera.

This is an impressive piece of genomic and bioinformatic research but I think the conclusion goes beyond the data. I suggest two ways to strengthen this paper. First, functional genomic analysis to establish causality. This was done for the honeybee way back in 2003, with RNAi. An alternative is to do additional comparative genomic analyses. There are numerous genomes of bees and ants available, going beyond the few consulted in the present manuscript. Either or both would strengthen the ability to make the desired conclusion, especially for a top-tier journal like PLOS Biology.

---

## [Editor Report · Decision Letter 2]

9 Oct 2025

Dear Matt,

Thank you for the submission of your revised Short Report "Genetic mapping in the red mason bee implicates ANTSR as an ancient sex-determining locus in bees and ants" for publication in PLOS Biology. On behalf of my colleagues and the Academic Editor, Chris Jiggins, I'm pleased to say that we can in principle accept your manuscript for publication, provided you address any remaining formatting and reporting issues. These will be detailed in an email you should receive within 2-3 business days from our colleagues in the journal operations team; no action is required from you until then. Please note that we will not be able to formally accept your manuscript and schedule it for publication until you have completed any requested changes.

IMPORTANT: I've asked my colleagues to include the following request among their own: Many thanks for providing all of your data and code in the Zenodo deposition. However, you need to include the Zenodo DOI in each Figure legend alongside the data file name. For example, the citation in the Fig 2 legend should be “Data from panels A-E can be found in files Data S01-S03 within DOI:10.5281/zenodo.17135872 and Supplementary Table S5.” (etc.)

Sincerely, 

Roli

Senior Editor

PLOS Biology

rroberts@plos.org